# A Framework for User-Focused Electronic Health Record System Leveraging Hyperledger Fabric

Mandla Ndzimakhwe [1], Arnesh Telukdarie [1,*], Inderasan Munien [1,*], Andre Vermeulen [1], Uche K. Chude-Okonkwo [2] and Simon P. Philbin [3,*]

[1] Johannesburg Business School, University of Johannesburg, Johannesburg 2006, South Africa
[2] Institute for Intelligent Systems, University of Johannesburg, Johannesburg 2006, South Africa
[3] School of Engineering, London South Bank University, 103 Borough Road, London SE1 0AA, UK
[*] Correspondence: arnesht@uj.ac.za (A.T.); inderasanm@uj.ac.za (I.M.); philbins@lsbu.ac.uk (S.P.P.)

**Abstract:** This research study aims to examine the possibilities of Hyperledger Fabric (HLF) in the healthcare sector. The study addresses the gap in the knowledge base through developing customization techniques to enable the simplicity and efficacy of Electronic Medical Records (EMR) adoption for healthcare industry applications. The focus of this research explores methods of using blockchain technology that prioritise users. The investigation of several concepts used in developing web applications has been determined. The study identified that an open-source project, known as Hyperledger Fabric, can be utilised to construct a novel method of storing EMRs. The framework provides a test network that can be customised to satisfy the need of several projects, including storing medical records. This research additionally outlines the difficulties encountered and problems that need to be resolved before Hyperledger Fabric can be successfully implemented in healthcare systems. Considering all types of blockchains available, the needs are met by Hyperledger Fabric, which offers a distributed and secure environment for healthcare systems. Blockchain has the potential to transform healthcare by putting the patient at the centre of the system and enhancing health data protection and interoperability. Also, by using grant and revoke access mechanisms, patients have complete control over their medical information as well as authorized doctors who are allowed to view records. This functionality is made possible by the chaincode defined in the blockchain platform. The research study has both practitioner and research implications for the development of secure blockchain-based EMRs.

**Keywords:** healthcare; blockchain; distributed network; ledgers; chaincode; hyperledger fabric network; grant access; revoke access; electronic medical record; electronic health record

## 1. Introduction

The achievement of universal health coverage is one of the top priorities of the World Health Organization (WHO) agenda for 2030 [1,2]. The ultimate objective of universal health coverage is to ensure every person around the world is able to access and afford quality healthcare regardless of where they are domicile. In joint efforts with the World Health Organization (WHO), South Africa and other global nations are planning the implementation of the National Health Insurance (NHI) scheme, which is intended towards enhancing the accessibility of quality healthcare services for all citizens [2]. To achieve a smooth transition and reach the identified end goal, the NHI is required to establish an electronic health record (EHR) system to register and track patients who visit several healthcare providers. EHR consists of patient health-sensitive data for diagnosis and medication [3]. This information can be distributed and become available within different health organizations. EHR systems are used to implement the electronic medical records (EMR) to organise patients' health data and information [4,5]. Treatment, medical diagnoses history, allergies, and test data are all stored in an EMR [6]. Generally, EMRs are useful

for healthcare organizations and their patients, yet many EMR management systems have issues that cannot be overlooked [7]. Indeed, the healthcare organization's EMRs are mostly stored in the local database in the traditional EMR management system [8]. Furthermore, although there has been widespread adoption of EMR systems in many countries, challenges remain, such as a lack of engagement with advanced functions of the system [9]. As a result, the patient's private information is exposed to dangers, such as deliberate tampering, unintentional data loss, and cyber-attacks [10].

Presently, it is often the norm that EMRs are managed by healthcare organizations from an access point of view. Moreover, ordinary patients have no privilege to take charge of their EMR. Patients do not have control over which parties can conditionally access and potentially use their information. In terms of data sharing between hospitals and patients, EMRs are distributed among healthcare organizations and the different healthcare organizations have different standards for their EMRs. When this happens, it can be extremely difficult for patients to access fragmented health records from several sources into a single chain at the appropriate time, especially when they are in a critical condition [11]. These EMRs contain private and public information for both healthcare organizations and patients, which is another factor that makes it difficult to share among peers.

To resolve such critical problems, this research study develops and adopts a general blockchain technology framework that prioritizes users; and is established through an advanced evolving open-source project referred to as Hyperledger Fabric (HLF) [12]. Further, this study seeks to contribute insights towards the knowledge gap by providing the best possible customization techniques to enable simplicity and efficacy of EMR adoption mainly for healthcare industry applications. The main limitation of the study is the focus on strictly developing a prototype that mimics the working application on the production level. The system is designed to store patients' information, grant access, as well as the capability for access to be revoked when the service is no longer needed by patients from their doctor. Furthermore, the application presented focuses on the aforementioned features to store and access medical information.

The layout of this article is structured as the following: Section 2 discusses the systems that have been used in medical record management in healthcare utilizing different types of blockchain technologies. Section 3 introduces the novel method; Section 4 outlines and shows an implementation of the prototype; and Section 5 discusses the results of the web development of the prototype. Section 6 provides the conclusions together with future works.

## 2. Literature Review

### 2.1. Electronic Health Records

Over history, the governance of South Africa's health system has been chaotic and fragmented, and its resources have often been poorly managed [13]. This has led to a system that is highly inequitable, expensive, and inefficient. Health records have historically been kept in paper format, kept in files that were sectioned off based on the type of record, with only single copies being made. In the post-apartheid period, the constitution conceived a deliberately decentralized management system strategy [14]. Increased efficiency, local innovation, empowerment, and accountability in the communities served as the goals of this decentralized management approach.

The basis for the development of the electronic health record (EHR) was developed by computer technologies in the 1960s and 1970s [15]. An EHR can be considered as a digital representation of a patient's medical records. In addition to making patient medical records more readable and accessible from different locations around the world, the usage of EHRs has further changed the format of medical records, which has had an impact on healthcare. The adoption of EHR systems came with challenges of high-cost implication and data entry errors as well as cybersecurity [16]. Resultantly, EHRs are often used as a complement to the paper format record system and not as a replacement. The application of EHRs in the healthcare industry offers a way to maintain patient safety and overall healthcare delivery,

while improving the efficiency of information sharing. However, there remains a risk of privacy violation and identity theft.

## 2.2. Blockchain

At the emergence of Industry 4.0 (4IR) technologies [17], blockchain has been the most promising technology to address challenges of security and risk management [18]. Blockchain is defined as a decentralized technology system that is used in several applications, namely the internet of things (IoT), finance applications, logistics, healthcare, and supply chain management [19]. Blockchain is an example of a distributed ledger technology (DLT), which is based on a peer-to-peer system for decentralised recording of transactions between different parties simultaneously across multiple locations [20]. Blockchain is considered a highly efficient method in cases where various participants have the right to access and interact with the mutual database [21]. In the healthcare sector, the decentralized application enables security, access and transparent exchange, and usage of medical data [22]. There are two different categories of blockchain known as public and private blockchain [21]. One typical example of a public blockchain is based on Ethereum, while a private blockchain is based on HLF (Hyperledger Fabric). The publicly available blockchain presented several challenges, such as inefficient execution, unauthorized access, and the system being expensive to execute. Since then, the private blockchain has been developed to make up for the identified drawbacks [23].

A private blockchain is also known as a permission-based blockchain system [24]. It has been developed among peers of the network by using basic structures called certificate authorities (CA) and membership service providers (MSP). HLF provides the benefit of a secure database and a consortium that does not need approval by all organizations and peers across the network [12]. The private blockchain also comes with the addition of an identity authentication mechanism. The identity authentication mechanism provides access only to the users who are authorized and through such an approach the system prioritizes enhanced privacy protection. In the system-shared database, data are stored with features such as revoked access and are traceable if the system is altered [12].

## 2.3. The Adoption of Blockchain Technologies in the Healthcare Sector

In recent literature, distributed technology has been viewed to be the most efficient and effective method to solve healthcare challenges [25]. Indeed, several studies proposed blockchain mechanisms for healthcare data management of medical records [12]. Private blockchain, which is also known as proof of ownership, proposed a secure ecosystem of sharing EMRs and mitigated evaluating processes [26]. The work produced quality information that can be targeted and successfully implemented in the exchange of medical data. However, due to the lack of global exploration and scaling mechanisms for the exchange of medical information, the system did not fully achieve a smooth adoption. Private blockchain for EHRs and personal health records (PHRs) contributed to blockchain solutions for confidential control in sharing healthcare data [27]. The disadvantage of this system is its lack of scalability and limited EMR data. Furthermore, public key infrastructure (PKI) is the famous cryptography protocol method of blockchain, which has been featured in the EHRs for the access of stored health information [28].

The PKI method is not based on the perspective of considering the access control and integrity of data kept in the blockchain. This technology encompasses the retrieval of health records and the exchange of the records between healthcare stakeholders. The approach of consortium blockchain was also developed to keep PHR and sharing of healthcare-related information within healthcare enterprises [29]. To adequately realize the proposed objectives focused on data sharing and usage, this study excludes factors such as data integrity, privacy, interoperability, and access control. Public blockchain that utilizes Ethereum for providing evidence to the network and hence access, uses EHRs as a secure, scalable solution for recording and keeping health information. This solution is one of the first methods that was adopted in the healthcare sector for storing medical

information [30,31]. The public blockchain method failed to copy the shared mechanism to promote transparency across all network parties and proof of integrity. Moreover, the Ethereum platform has provided the advantage of a cost-effective model but failed to pledge scalability [32]. Despite the age of the inception of Ethereum, the solution cannot provide a replacement for current methods as well as establish capabilities and privacy as it uses cloud technology [29]. The security- and performance-based models demonstrated improved utility. In the domain of healthcare data storage, exchanging information and interoperability are the major difficulties that need to be solved by utilizing blockchain-enabled solutions.

### 2.4. User-Focused EMR Systems Using Blockchain

In many EMR systems, patients are treated differently when compared to other stake-holders, such that they have least rights to access their health information, although the whole system is built around their information [33]. Indigo is one of the systems that have been created to address and allow patients to keep and manage a copy of their EMRs [34]. The main objective of the system was to empower patients and give them privilege to control the medical records. Most of the user-centred systems focus on access control, and by that privacy is viewed as precaution [34]. In such systems, the attention is on the regulations and policies on parties that can access data. However, this system focuses on different aspects such as giving patients features to grant access or deny access if medical service or health attention is no longer needed by information owners, which are the patients.

The techniques and methods are detailed and discussed in Section 3 from the analysis to the implementation of the prototype. In this work, the healthcare network direct peers' identities are shared among the network to ensure accountability; for example, for a medical practitioner to access patient information, it is up to the patients to give them their IDs to provide access. By doing this the system will ensure that all the actions that are involved reflect back to the patients. In recent works the HLF contains many mechanisms and has an intricate structure [35]. These mechanisms can be used in several various ways. This work also seeks to find the most efficient and best application to better develop the EMR system differently compared to existing scenarios.

### 3. Materials and Methods

The summative block diagram in Figure 1 illustrates the adopted methodology for developing a general blockchain technology framework that prioritizes users. The method is established through an advanced evolving open-source project herein referred to as Hyperledger Fabric (HLF) [12].

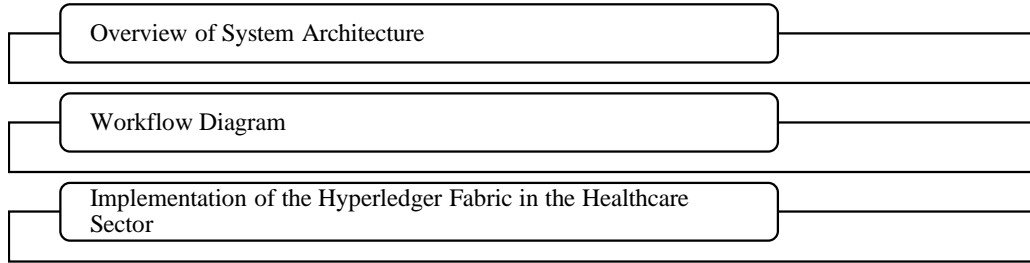

**Figure 1.** Summative steps of the research methodology.

### 3.1. Overview of System Architecture

Figure 2 illustrates a diagrammatic overview of the architectural setting of technology used to develop an application by adapting the HLF test network to the healthcare network. The blockchain operator is required to set up the initial configuration of the network and grant the required access and credentials to users who control the system. The orderer certificate authority (CA) is created in the docker fabric image and utilised by all the peers of hospitals. The orderer acts as an admin that approves all organizations and validates the

credentials of their peers. All the elements of the application are connected to the network in a communicative manner. The backend code and smart contract are written in JavaScript, and ExpressJS serves as the REST API server. Angular 11 framework is utilised to create the user interface to enable a user-friendly experience. REST API calls are used for frontend to backend communication, and JSON web tokens are used for authentication. Official frameworks grant developers the flexibility to choose and use their strongest programming language to develop chaincodes or backend codes to configure the access of information by the network participants. The available languages in their latest version, which is 2.x, include Java, Golang, JavaScript, and Typescript.

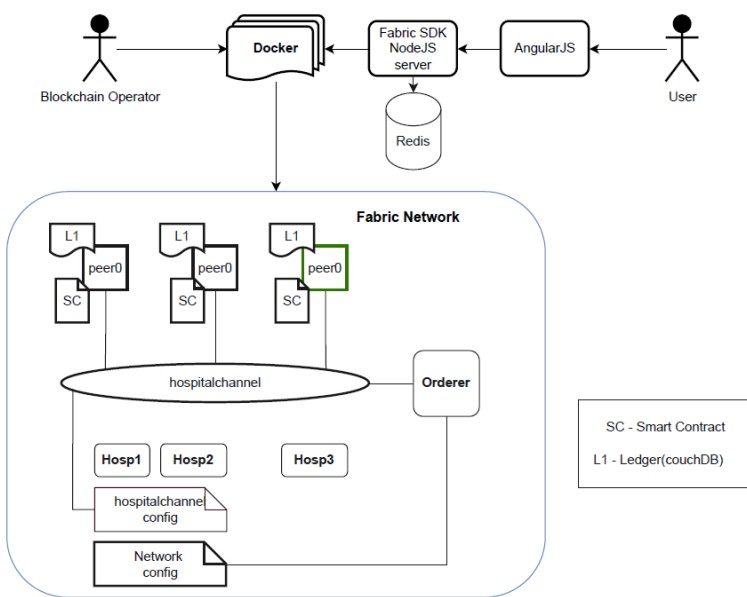

**Figure 2.** Overview of System Architecture.

For the system design, two hospitals are connected at a single channel called the "hospital channel". When the network is operational and connected via the same channel, it is possible to add additional hospitals. The framework further integrates the data modelling preferences of LevelDB and CouchDB. CouchDB is considered the solution due to its flexibility. Blockchain can store information, but those systems are not reliable and not scalable as they are slow. CouchDB is adopted for its file management capabilities and the ability to handle metadata, unlike LevelDB. In this scenario, CouchDB keeps the world state information. Ledger is used as the basis of transaction logs and world state. All transactions are stored and accessible in the transaction log starting from the genesis block. Another important element is Redis, which has a key valueDB for the main purposes of storing the doctor's credentials.

*3.2. Workflow*

Figure 3 provides a simple schematic of the workflow of the entire proposed system. The system has three personas that are divided into the categories of administrator, patient, and doctor, with features as illustrated in Figures 4–6, respectively. Every hospital has a user interface where administrators make the required modifications regarding patients.

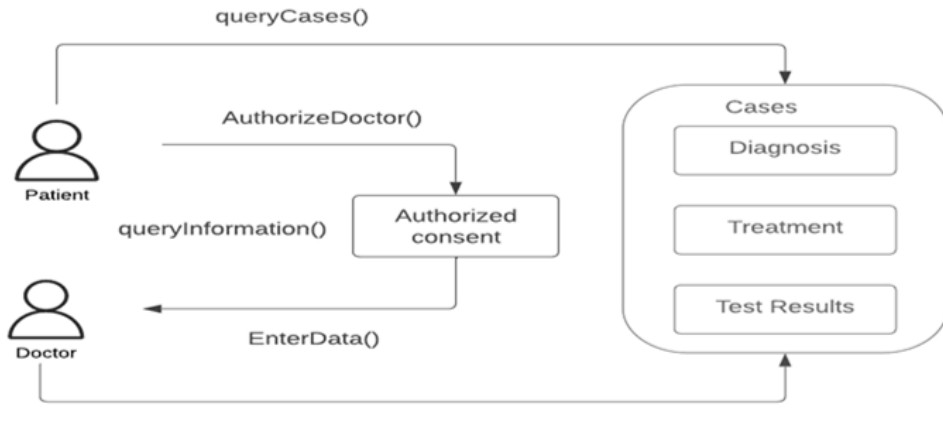

**Figure 3.** System workflow.

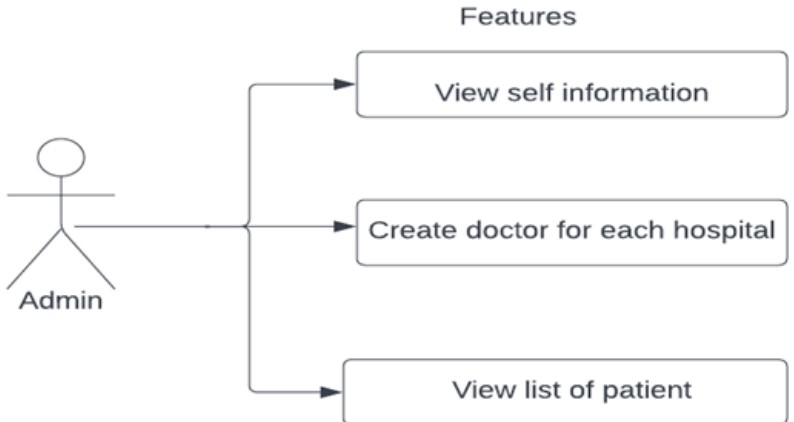

**Figure 4.** Illustration of the admin persona (or features).

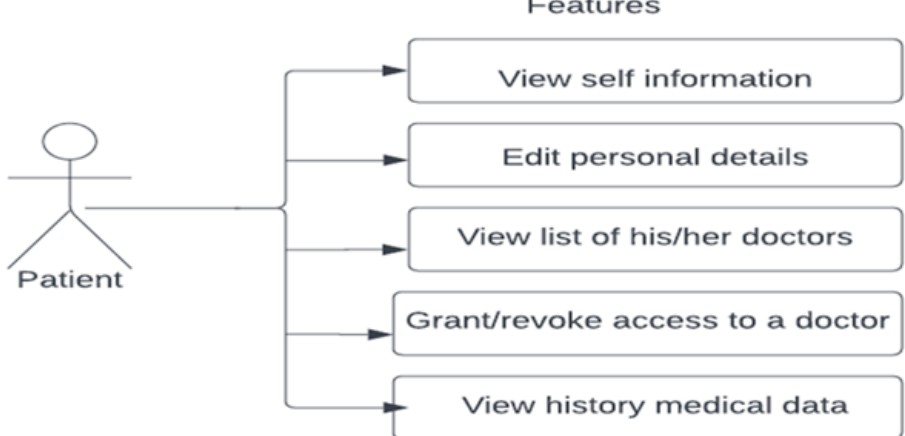

**Figure 5.** Illustration of the patient persona.

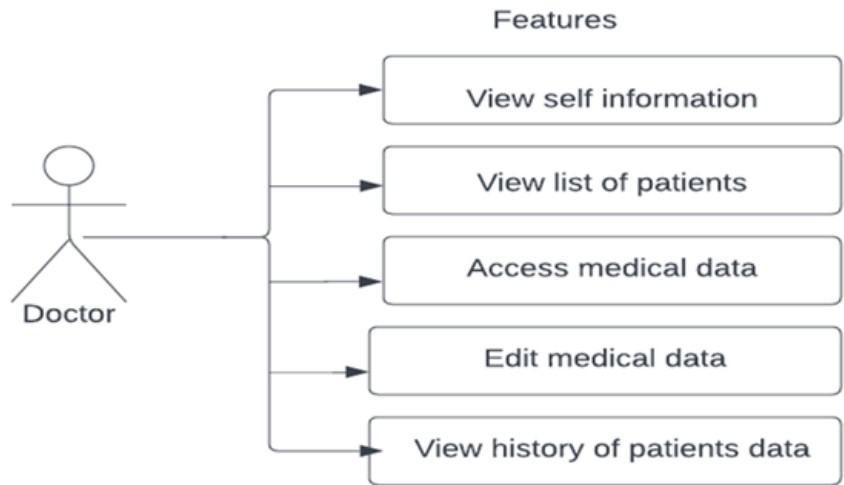

**Figure 6.** Illustration of the doctor persona.

*3.3. Implementation of the Hyperledger Fabric in the Healthcare Sector*

3.3.1. Technology Stack

To implement an end-to-end blockchain system using HLF, various components are involved in the workflow. The following components are used in the development of the framework:

- Hyperledger Fabric: These are smart contracts built inside this blockchain [12].
- Docker Compose: Used to deliver software packages called containers (docker-compose-ca.yaml, docker-compose-couch.yaml, docker-compose-net.yaml) [12].
- Couch DB: Open-source database, which allows the storage of data in JSON format and is used as an external state of a database for Hyperledger fabric [12].
- Node JS: This is an open-source cross-platform backend where the script runs in the terminal and executes JavaScript code outside of the browser. In this project, Node JS is used to provide API to react with Hyperledger Fabric blockchain (which performs the first level of user authentication and acts as the gateway to the Fabric smart contracts) [12].
- Angular JavaScript: This framework has been used to build the client application web interface.
- Interplanetary file system (IPFS): This is a peer-to-peer file storage network for storing and sharing data in a distributed file system. Content is accessible through peers located anywhere in the world [12].

Based on a combination of the aforementioned information on implementing an end-to-end blockchain system using HLF, Figure 7 demonstrates the interconnection of the adopted technology when a user is using the system and sending different requests.

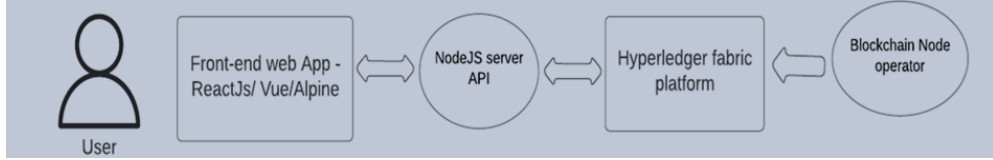

**Figure 7.** Schematic technology stack.

3.3.2. Healthcare Network

The framework of the system is the HLF network, comprising hospitals engaging in the distributed ledger channel with their peers. All hospitals are connected to a single channel, as shown in Figure 2. The patient data management system (PDMS) is customized using the test network provided by HLF. Also, by updating the docker files, configuration

files, and accompanying certificates, the companies using the current network are modified to represent the hospitals. The key modifications are to the organization names in the configtx.yaml file and the accompanying CAs in the docker-compose file to add the hospital. As illustrated in Figure 2, two hospitals are added to the upgraded network along with a channel, whereby all required credentials for the company and peers participating are generated.

### 3.3.3. State of Distributed Database

According to the official HLF documentation, LevelDB and CouchDB are the two peer databases that HLF supports. In contrast to CouchDB, which stores data as JSON documents, LevelDB is the HLF database by default and stores data as key-value pairs. Compared to LevelDB, which only supports composite key queries, CouchDB supports rich queries over JSON documents for both medical and personal information [12]. For this reason, CouchDB is utilised as a peer database to model the patient health record of each patient as a JSON datatype. A patient ID field, one of many that are present in every EHR, is used to identify the owner of the record [36]. Only the doctors listed in the list are permitted access to this data; otherwise, HLF rejects access.

Another field is an array of a list of doctors authorized to access the record. CouchDB is used so that grouped queries are executed. CouchDB indexing enables the grouping of JSON documents (in this example, health records) based on any field that is available in the JSON document [24]. No indexing or design document (DDOC) features of CouchDB have been used yet, although they can be considered in the future. The number of CouchDB Docker images depends on the number of peers and runs on the same server as the peer. Since each peer has its ledger, an HLF network requires a single CouchDB image for each peer.

### 3.3.4. Deployment of Chaincode to the Network

Contracts are used to implement all the executable business logic of the application; therefore, smart contracts are used to execute assets such as create, read, update, or delete activities on the distributed ledger [12]. Depending on the architecture and programming language used, chaincodes can be various functions or even different files (or classes). In this case, smart contracts have been implemented using JavaScript. One function has been written for each capability of the proposed system to match the interface with the HLF network to maintain the simplicity and modularity of the architecture.

The entity on which network transactions are expected to take place is often the focus of smart contract development [35]. This is the EHR in this situation, and smart contracts are created around it. When an admin registers a new patient, the contract CreateRecord() assists in creating a new EHR in the distributed ledger. The contracts UpdatePatientInfo() and UpdateRecord() are used to update the personal and medical information of the patient, respectively. When a doctor or patient wants to read an EHR, the doctorReadRecord() and patientReadrecord() contracts are triggered. To retrieve the history of a certain EHR, the GetRecordHistory() contract is used.

The HLF blockchain network has a function called GetHistory, which allows users to retrieve the history of transactions that have taken place on a certain entity, and is advantageous since the global state simply keeps track of the most recent or updated state of a record. The history feature allows for the tracking of any earlier transactions. One of the fundamental components of the PDMS is the ability to grant and revoke access, which is accomplished using the GrantAccess() and RevokeAccess() methods. This enables the patient to approve or remove a doctor's access to their electronic health record. Since it is possible to regulate access to the data at the time of data retrieval, it is implemented in smart contracts so that the data is present in the fabric network before the check is finished. This is accomplished by having a list of doctor IDs who are permitted access to the EHR in an access control list called DoctorAuthorizationList within the EHR.

The medical doctor's ID will be added to the patient's EHR when the patient specifies the doctor to whom access must be provided through the user interface. A patient can choose a doctor, and that doctor's ID will be removed from the DoctorAuthorizationList, just as how a doctor can have access revoked. This list and the way to give or withdraw access will only be accessible to the patient; neither the doctor nor anyone else will have access to it. Additionally, the doctor will not be able to see this list. In the system, if a doctor attempts to interact with a patient's electronic health record (EHR), the system checks the doctor's user ID against the DoctorAuthorizationList in that EHR. If the doctor's ID is not present in the list, the system will inform the doctor that access is denied and will not allow them to view or modify the patient's EHR. Like this, the doctor is permitted to examine or alter the EHR if the ID is available. Typically, smart contracts are implemented on the blockchain network after being packed into chaincode. Therefore, a chaincode may include several smart contracts, and when this chaincode is installed on the network, all the contracts are made accessible to the application. There are four processes involved in deploying chaincode to the HLF network [37]: packaging the chaincode; peer installation of chaincode; approving a chaincode definition for a channel; and committing a chaincode definition. Once the HLF network is operational, all four processes may be carried out simultaneously by using the deployCC command. The route of our chaincode and the language in which it is written must both be passed with the appropriate flags.

### 3.3.5. Use of Software Development Kit (sdk)

Hyperledger Fabric Client SDK offers APIs (application programming interfaces) for interacting with smart contracts, adding transactions to a ledger, and querying the ledger [38]. The following packages are available via the Fabric SDK:

### Fabric-Ca-Client

The fabric-ca offers APIs for participants (i.e., admin, patient, and doctor) to sign up and enrol to create trustworthy identities on the blockchain network. The package develops a new CA client that can communicate with the hospital's CA server to enrol and register participants.

### Fabric-Network

The APIs needed to connect to the Fabric network, submit transactions for queries, or alter the ledger are included in this package. Gives access to APIs that may be used to manage the wallet that is used to maintain identities and to build a connection profile using the connection profile JSON that is produced when a CA is formed.

### Fabric-Common

It encapsulates the standard code that is used by all fabric-SDK-node packages to submit transaction invocations to the Fabric network. It offers APIs for event monitoring, logging, environment variable configuration, program arguments, and in-memory settings.

### Wallet

The wallet serves as an identity and is a crucial component of the Hyperledger Fabric SDK, as was already explained. It is used to save the Fabric metadata, the private key in an identity file that has been authorized by a certificate authority, and the public key. Wallets come in a variety of forms, including file, in-memory, and database wallets. The file type is used in the creation of the patient data management system. The Gateway class utilizes the mspID and the type stored in the user's wallet during connection formation to connect to the network and check the user's access privileges to the channel in question [39].

### Use of the API

The Gateway class, included in the fabric network package, is the primary class that enables the communication between the Fabric SDK and the network. When an assert has

been performed, a gateway or link is established to a peer or user inside the blockchain network, allowing access to the chaincode and channels.

The registerUser() function in the SDK is called when the Register Doctor/Patient action is executed, and it then collects the user's information, including their ID, username, and the hospital they will be linked with. A wallet is then generated and an identity file for that user is added to it after obtaining the network parameters for that hospital. This acts as the user's identification while attempting to enter the Fabric network. The hospital's administrative staff must complete this procedure, and they must also register as administrators in the same way users must first register. When a patient registers, the smart contract's CreateRecord() method is called, which creates a new record for that patient with the necessary personal data. The user will have access to the SDK's APIs once he or she has registered. To prevent each user from manipulating the data, some APIs are only available to specific users. That is, only the hospital administration may register a doctor or patient.

Updating patient health records, on the other hand, is only available to doctors because it updates the EHR's diagnosis and treatment. Furthermore, only patients have access to update patient personal information and grant or revoke access to doctors since doctors are not permitted to add personal information or change the DoctorAuthorizationList. Doctors and patients who have access can see a patient's data and read their medical history, since this is how the EHR is accessed and read. Each person has a role (admin, doctor, or patient) that they may utilize to prohibit certain actions or trigger specific smart contracts depending on their role.

Implementation of JSON Web Tokens

JSON Web Tokens (JWT) have been included to manage API permission and to keep the user's session active. By doing this, it is possible to ensure that the person who logged in also accesses the API. To save the server from having to retain the session information itself, a token is produced when the user signs into the application using the username and password of the user. This token is then signed off, encrypted, transmitted back, and kept on the client side. The JWT checks the token with the key (the user's password) the next time the user uses the same token to access the API, and if it has not been tampered with or it is the same user, it grants access to the API. To maintain the focus on Hyperledger Fabric rather than application development, the user login credentials are kept in a file as JSON and are checked against.

3.3.6. Client/Front-End Development

Login Screen for Admin

Each hospital that joins the network receives an admin. When adding the hospital to the network, administrative information must be present. The appropriate hospital CA configuration's fabric-ca-server-config.yaml file contains the admin information (username and password). Since the online application is the same for all users, the admin must select the hospital and input their credentials on the login screen, which is seen in Figure 8. These credentials are compared to the ones set in the Hospital CA configuration YAML file and saved on the blockchain network.

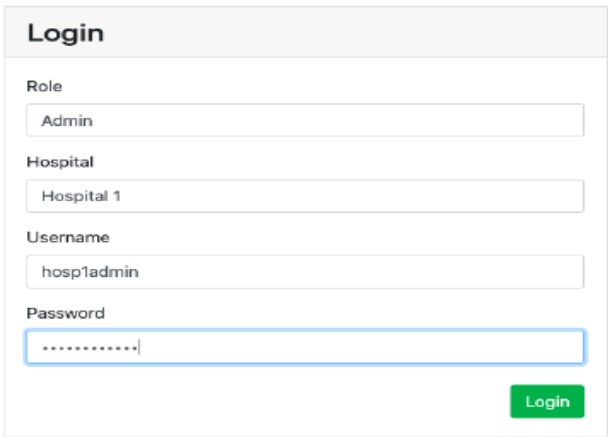

**Figure 8.** Admin login.

Admin Dashboard Screen

Once the admin signs into the system, a dashboard is shown, which includes a list of every patient in the network as well as the features shown in Figure 9. A transaction is established in the ledger to obtain all the patient objects in the global state when the admin contract, which is used to retrieve the patient list, is invoked. The contract limits the access by admins to patient information just by their names.

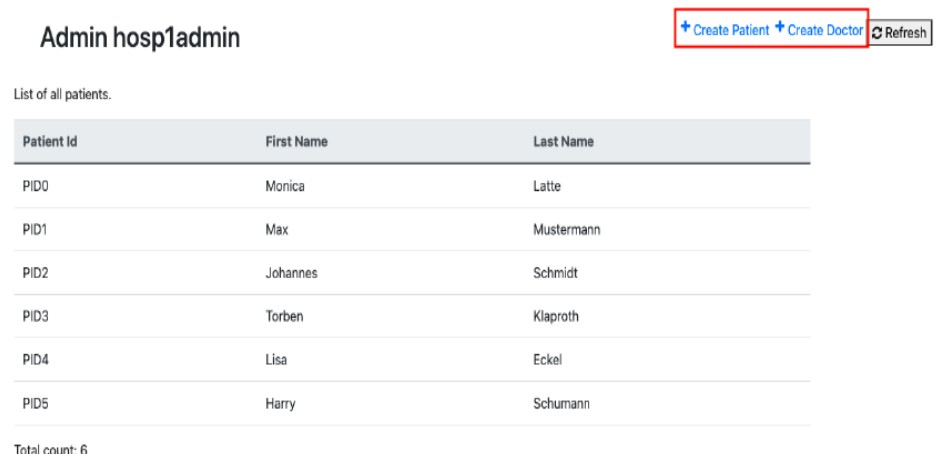

**Figure 9.** Admin dashboard.

Asset-Create-Patient

An item in the world state can be created by the administrator by generating a patient who creates a transaction in the ledger. A transaction is then established in the ledger and sent to all network peers. The patient data is checked for accuracy, approved, and then kept in the ledger. The patient is also generated in the network as a client, and by utilizing this client they may communicate with the ledger. Additionally, after saving, the patient receives temporary login information that must be changed the first time they log in for security reasons.

Asset-Create-Doctor

Doctors can be created by the admin. Since the doctor is not a ledger object, a client is made on the network so that the doctor may communicate with the ledger. Upon registration of the doctor, there is no relationship with the ledger.

Patient Login

On the patient's initial visit, the hospital gives him or her a temporary password and username. When a patient logs in for the first time, the notification asking for an instant change of the temporary password appears after the patient selects the role of the patient and enters the login and password issued by the hospital. As seen in Figure 7, the patient must input a new password of his or her choosing. After this, the new password is hashed and kept in the ledger. The patient only has to choose their position and input their new password and username going forward. The temporary password as well as the new password are hashed and kept in the ledger. Every time a user logs in, the password they input is hashed and matched to the hash value of the password that is kept in the ledger.

View Patient Information

The patient can access both his or her personal and medical information after successfully logging in. The patient's information is obtained from the global state by using the patient contract.

Feature Grant/Revoke Access

Access to and from the doctor may be granted or revoked by the patient. When a patient permits access to a doctor, the doctor is the only one who has access to the patient's medical information and medical history. If a patient revokes access, the doctor no longer has access to the patient's medical data. The screen that shows the functions or features to grant access to the patient's doctors in case they need them or revoke access from some doctors is provided in Figure 10.

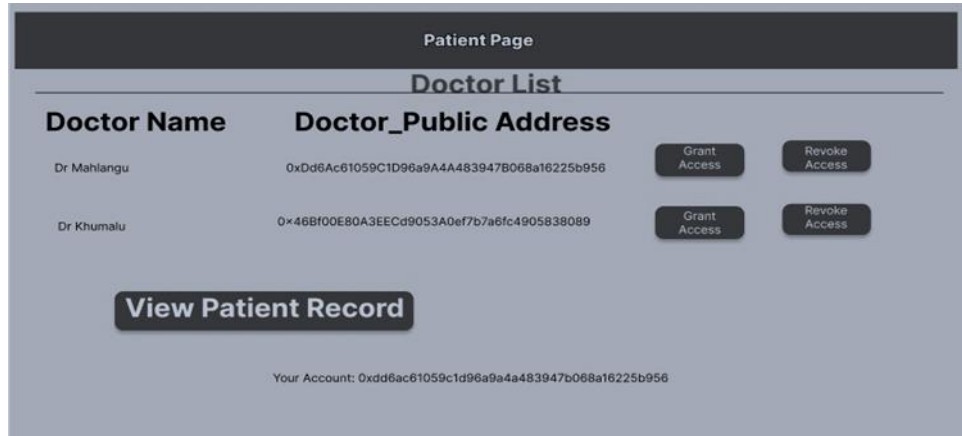

**Figure 10.** Grant or revoke feature.

Update Personal Details

The patient has a right to change his or her personal information. If there are any adjustments, the patient contract is used to update the ledger with the new information.

View History

The patient has access to all personal and medical records dating back to their first appointment with each doctor. The availability of the getHistoryForKey API in the Hyperledger fabric, which retrieves the patient's history, makes this capability feasible. The patient is given total access and transparency over all transactions to this system, which highlights the data transparency of the system implementation (see Figure 11).

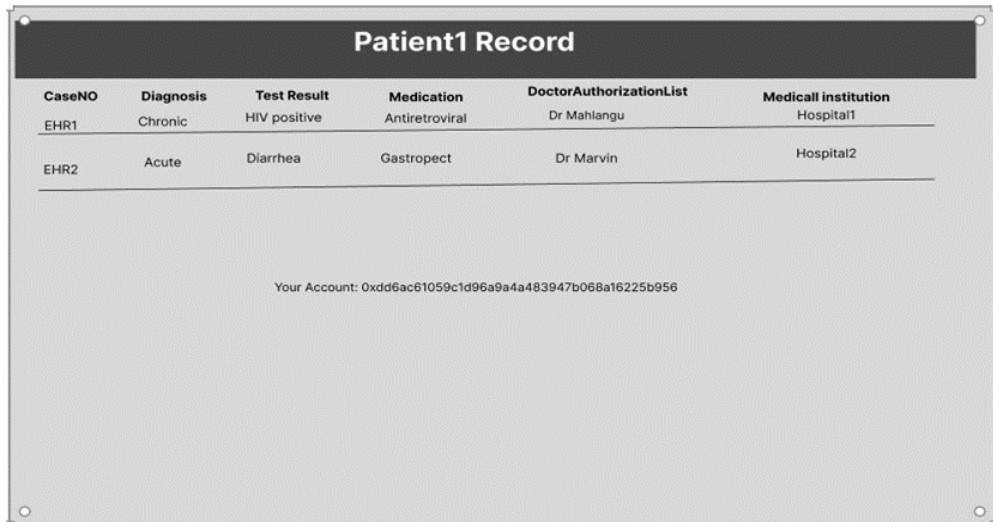

**Figure 11.** History of patient medical records.

Doctor Login Screen

Upon login, a doctor must choose the role of "Doctor," the hospital to which they belong, and the appropriate credentials on the login screen. When a doctor successfully signs in, their information appears alongside patient information.

Feature to View Patients

If a patient grants access, the doctor can access the list; otherwise, the list will be empty. The list has the same appearance as the administrator's dashboard, but since the doctor may see patient data, each item has a view more button. This takes the user to the patient information page, but the doctor can only see the pertinent fields. Here, the doctor may view the patient's most recent information and condition.

Update Medical Information

A doctor can treat a patient by altering their medical information. The patient information page is displayed after clicking the save button. Figure 12 shows the possible features their patients will have and the transparency of the system.

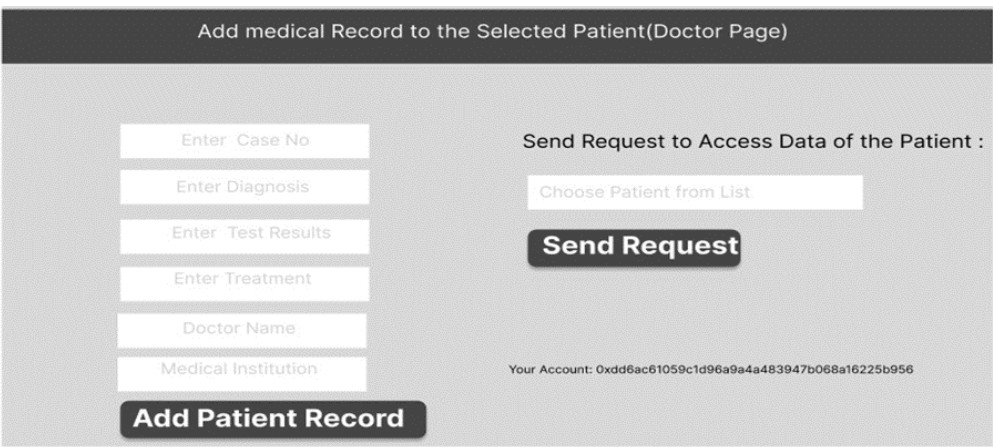

**Figure 12.** Update patient record.

Medical History

The patient's whole medical history is available to the doctor, who may use this information to have an improved understanding of the patient's condition and determine how to administer an appropriate course of treatment or medication. The same constrained fields are displayed in patient details. Additionally, information that has been updated and altered is useful for understanding the course of treatment.

## 4. Results

In the course of developing this system and implementing this framework, several factors have been discovered and analysed. These factors are not only related to the personalized medical app that was developed but also identified during the use of blockchain technology and from all the open-source projects that were utilized in the creation of the app. The Appendix A section shows the shows the high level of developing core components of the healthcare network application which is mainly scripts for the run network as well as the development of the user interface.

### 4.1. Pros and Cons Analysis of the Fabric Network

4.1.1. Pros

The option to add plugins for the consensus method and identity management is provided by the fabric architecture. This makes it easier for businesses to integrate their identity management systems with fabric. The confidentiality of data is the primary goal of businesses adopting permission-based blockchains. The MSP component of the fabric offers user authentication, while also issuing and validating certificates. Data is secured since all the participants are known in the blockchain network. This considerably improves the fabric's performance. A blockchain network can have a private channel created for a selected few users using Fabric. Although there may be a small number of transactions that should only be seen by specific individuals, this mechanism may be adopted to ensure privacy.

4.1.2. Cons

The architecture of Hyperledger Fabric is quite intricate and Fabric currently supports CouchDB and LevelDb. When MRI reports and high-resolution images are needed for storage in the healthcare industry, fabric offers limited database support. Again, it is still a challenge to host documents or big data in the blockchain database, which results in the use of IPFS, that is, a publicly distributed platform. However, the problem is when blockchain is used as a database, the whole system becomes slow and not user-friendly for the application users, in this case, doctors and patients.

### 4.2. Challenges

The objective of this research project was to design a system with careful consideration of the patient-focused approach and to ensure that the flow between organizations is efficient. Before any preliminary requirements were captured, the system design itself has proven to be difficult.

Keeping patients at the centre of the design, considering their needs, and incorporating different channels for safe communication between hospitals are all required to ensure the security of the EHR. Having a channel for each hospital led to the issue of linearly expanding channels, which puts a strain on the system and adds to the complexity of both system implementation and maintenance. Consequently, private data collection is utilized as a straightforward substitute for many channels to prevent this situation.

Security implementation for patient data in the ledger on a peer level is unfortunately fraught with significant difficulties. The fabric problems outlined in the previous section make it impossible for the private data collection technique to properly function. The method fails to be implemented for data re-encryption for two main reasons. The first issue is the lack of an effective re-encryption module in NodeJS. Some of the libraries are listed

in [40]. However, none of them was employed here. One works, but it does not suit the way a typical system generates private and public keys, and the library cannot comprehend the format.

Although it takes keys in the normal format and supports public-key cryptography, Node-RSA does not support re-encryption. The chance to comprehend and build a re-encryption algorithm is a time-consuming but effective strategy to adopt. A further issue is the generation of a user's certificate and private key. However, it was not possible to locate a hash key in the fabric framework. To identify this problem and make the re-encryption technique work, further investigations are needed.

The Hyperledger Fabric network faces some difficulties because of the implementation of private data collection. The first difficulty arises when a patient transfers hospitals and their wallet information is added to a new collection of records at a different hospital. Since the old wallet information contains the mspID of the old hospital and the user is not authorized to access the new collection, the information in the old wallet will be invalid. Therefore, the solution is to move the EHR to a new collection and dynamically swap the patient's wallet in the backend. The creation of a workaround for extracting transaction history—as was described earlier, owing to the lack of APIs to extract history—would then however provide another challenge. One fabric-peer container constantly shuts down when the network starts up for whatever reason. Additionally, not all peer configurations are incorporated when generating Common Connection Profile (CCP) files.

*4.3. Hyperledger Drawbacks*

As stated previously, Hyperledger is a blockchain platform that is open-source and still under development. Indeed, every day, improvements and new features are being created. The database that accompanies the adoption of the Hyperledger Fabric as a healthcare system is one of its biggest downsides. The databases in the state database that currently store data in JSON format are CouchDB and LevelDB. Medical reports sometimes include significant amounts of data, such as images from scans and other media files, which results in massive amounts of data and different problems with CouchDB or LevelDB. This can be avoided by storing the actual data in a cloud storage system or another database that enables the simple maintenance of big data sets, and storing the metadata in a CouchDB database, such as the URL to the actual data and the key values.

This approach would undoubtedly move the data from the distributed system and place it in a more centralized database, defeating the purpose of using Hyperledger Fabric once more [41]. However, the system would still be able to provide patients with simple access to their EHR. This method of managing patient data assumes that patients are among the parties involved, that hospitals exist to care for patients, and that the smart contracts implemented in each hospital comply with this assumption. Additionally, in a perfect world, institutions like insurance companies would be active in healthcare, even though they would need access to the EHR. It is expected that this prototype does not involve any such organizations. However, incorporating outside groups would significantly change how the system functions. As noted in the introduction, due to the size of the data involved, scalability may take a significant amount of work. For instance, Brazil's Unified Health System handled 1.4 billion patient visits in the country's healthcare system in 2018alone, and approximately seven billion patient visits took place in China in 2017 [41]. In such cases, infrastructure, security, and maintenance may need significant investment, both financially and physically. As mentioned in the difficulties section, there needs to be more development in Hyperledger Fabric's private data collection section to properly meet the needs of the healthcare domain.

Although the Hyperledger Fabric community is working to introduce a history index and chaincode API, such as public data history, to enable the query of a private data key's history, the fabric currently does not offer the API getHistoryForKey for a private data collection [42]. The only roles for an HLF Registrar in the current version of Hyperledger Fabric are client, admin, and peer. However, the problem with having only three roles

is that all clients now have the same set of permissions, but a blockchain allows for an unlimited variety of role types and permissions for each user-defined role. Instead of only the customer in the scenario, the user-defined roles can also be the doctor and the patient. In this case, patients and doctors each have their own set of guidelines for using clients to access user attributes.

The name and area of expertise of the doctor are kept in the attributes of the identity in this case because they are not assets to the ledger. Although the admin user can read the attributes, patients in the blockchain network are unable to do so because they lack access to view the attributes of the doctor. This problem emerges because the system is unable to apply a different set of permissions. The following Table 1 displays the summarized comparison performance analysis of the private and public blockchain. Further than these components used in analysis there are more elements that can be used to further precisely expand analysis. Private blockchain is the most promising method to promote privacy, particularly in the case of healthcare data, which is categorically critical data. However, the framework can be modified and simplified to accommodate and motivate everyone who has interest in leveraging it and coming up with smart, innovational applications to help the world at large.

**Table 1.** The summary of the comparison of public and private blockchain frameworks.

|  | **Public Blockchain (Ethereum)** | **Private Blockchain (HLF)** |
| --- | --- | --- |
| Smart contracts | Written in Solidity, Migration | Java, JavaScript, Golang |
| Consensus | proof-of-stake (PoS), proof-of-work (PoW) | Different approaches |
| Scalability | Low | Higher |
| Privacy | Transactions are private | Transactions are private |
| Cost | High | Low |
| Crypto | Mining Ethers | None |
| Transaction speed | 20 tps | Greater than 2000 tps |

### 4.4. Application Scalability

As stated previously, the limitation of this application was to develop a prototype that will mimic the real-world application. The design of the system has considered scalability. In the implementation, it can be seen that hospital 3 has been deliberately developed separately and merged to the network to show that more organisations can easily be developed in a script and join the network. This would apply in the development of a real application of an EMR system in a country with a large number of hospitals and other medical institutions. All these institutions should be part of the network so that when the patients want to use it, their data are easily distributed and available when they are required. Another aspect that is not considered is to look at real big data that is currently kept in hospitals and investigate the possibilities of transformation considering the data to this system.

### 4.5. Comparison of Database Systems

Databases often have a single point of failure and are centrally managed by database administrators [43]. Since a database administrator has complete control over the database, if that administrator's security is broken, the entire dataset in the database can be changed. In contrast, blockchain makes many copies of the data in the form of immutable distributed ledgers and decentralizes data storage [44]. The single point of failure is eliminated because each member of the network has their ledger and can view any changes made to the ledgers. As opposed to the blockchain, if one ledger is corrupted, it will be immediately fixed using a smart contract (i.e., the network's business logic) and consensus. Database managers may modify data in conventional databases [45]. When a new user joins the network and shares

data with existing users, all the shared data is instantaneously appended to each ledger and made available to all users at once.

Blockchain ledgers are immutable [46], which denotes that any data once deposited there will remain there indefinitely, and each small modification is recorded in all distributed ledgers. Conventional databases have CRUD (create, read, update, and remove) activities that allow users to update and delete existing data in addition to writing new data and reading all data [47]. Whereas, considering the immutability of distributed ledger technology in blockchain [48], only read and write activities are possible.

Therefore, every member may observe how the current state of the data was reached over time by looking at the appending history, and every participant knows that the data retrieved on his ledger is valid because copies of only valid data are appended in the blockchain.

## 5. Conclusions

The proposed and implemented architecture using HLF serves as a workable replacement for the current system, considering the complex patient–data management scenario as the primary motivation. The solution offered is based on Hyperledger Fabric to address the identified problems in the current data management systems and methodologies in the healthcare sector. The system's components can be modified to fit the design specifications. This allows simulating the scenarios according to how the actual application would behave, providing a wealth of information on the complexity involved in managing a patient's health record, as well as the difficulties and drawbacks in implementing the system itself in terms of ease of use, security, scalability, and maintainability. In addition to what has been accomplished with the prototype, consideration has been given to further theoretical ideas that, if put into practice, may further enhance the system in order to realize such gains as the decentralized blockchain approach would offer. Furthermore, a comparison has been made between a standard centralized database strategy and the decentralized blockchain approach.

This empirical study has identified that adopting a system based on HLF to manage medical data electronically is a workable approach. An EHR should be confidential, sensitive, and unchangeable. Due to its distributed, trusted, and immutable nature, Hyperledger Fabric can be used to directly provide these characteristics. Access to the medical history of patients is maintained on the blockchain through the transaction history recorded for the ledger, which adds a layer of protection. To provide the individual creating the data with an improved level of control, a patient-centric system can be created. Blockchain-based systems can reduce the drawbacks that arise from a single point of failure in databases [46]. Furthermore, a blockchain system can help to address some of the most pressing issues in modern healthcare, such as interoperability, but only when it is fully implemented, that is, at the state or even national level. Unfortunately, there are still certain unresolved problems that must be solved before this can be accomplished completely. As a young and developing technology, Hyperledger Fabric lacks certain features, such as the ability to obtain transaction history when private data collection is used. This may require extra databases outside of Hyperledger Fabric to maintain complicated data in addition to CouchDB and LevelDB. Additionally, there are also other performance difficulties including the low maximum number of transactions that can be processed in each amount of time.

In conclusion, it can be identified from this study that the personalized medical system is a practical and useful application of Hyperledger Fabric. However, there is room for improvement in this area within Hyperledger Fabric before a finished product can be developed, and that additional components or technologies outside of Hyperledger Fabric are required to make it successful.

## 6. Future Work

The following areas for future work are recommended:

- Currently, the metadata concept keeps the actual patient data on the blockchain network as developed. If such a system is to be used as a national or global solution, this may become impossible because the blockchain would become huge. Conversely, it may be preferable to merely keep the metadata for health records and accesses on the blockchain, with the actual data remaining in a conventional database. This hybrid method may make it possible to take advantage of some of the features of both traditional database systems and decentralized systems. Therefore, development work is required in this area.

- Although the project developed a prototype, the system was designed with scalability in mind. It is envisaged that many hospitals and other medical facilities will join the blockchain network for the implementation of this system. One organization may have multiple peers, and there may be different sub-channels and endorsement policies. Further studies are required to address this feature of the system.

- To protect the security of EHRs in the future, a workaround for private data collection utilizing a modified data structure needs to be developed. Additionally, retrieving transaction history from private data may prove to be a practical solution to the problem of acquiring transaction history if it is supported by the next version of Hyperledger Fabric.

- Multiple orderer systems are used in Hyperledger Fabric v2.x to develop a fail-safe system that is crash tolerant. For example, if there are three ordering nodes in a system and one node fails, the ordering service will continue to function using the other two ordering nodes and by electing a new leader. Raft is the initial step toward Fabric's creation of a byzantine fault tolerant (BFT) ordering service, according to official Hyperledger Fabric documentation. Consequently, further development work is required to enhance this functionality of the system.

- The system manages medical information and, normally, healthcare processes are incorporated with consultation. Therefore, a consultation feature needs to be developed to improve the standard of the app and make the process easier for both the doctor and patient.

**Author Contributions:** Conceptualization A.T., I.M., A.V. and U.K.C.-O.; methodology M.N.; software, M.N., U.K.C.-O. and A.V.; validation, M.N.; formal analysis, M.N.; investigation, M.N.; resources, M.N., A.V. and U.K.C.-O.; writing—M.N., A.T., I.M. and S.P.P.; review—A.T. and S.P.P.; visualization M.N.; supervision, A.V. and U.K.C.-O.; project administration, I.M., A.V. and U.K.C.-O.; funding acquisition, A.T. All authors have read and agreed to the published version of the manuscript.

**Funding:** This research received no funding.

**Data Availability Statement:** Not applicable.

**Conflicts of Interest:** The authors declare no conflict of interest.

## Appendix A. Network—Organization Setup

```yaml
networks:
  hospital:

services:

  ca_hosp1:
    image: hyperledger/fabric-ca:$IMAGE_TAG
    environment:
      - FABRIC_CA_HOME=/etc/hyperledger/fabric-ca-server
      - FABRIC_CA_SERVER_CA_NAME=ca-hosp1
      - FABRIC_CA_SERVER_TLS_ENABLED=true
      - FABRIC_CA_SERVER_PORT=7054
    ports:
      - "7054:7054"
    command: sh -c 'fabric-ca-server start -b hosp1admin:hosp1lithium -d'
    volumes:
      - ../organizations/fabric-ca/hosp1:/etc/hyperledger/fabric-ca-server
    container_name: ca_hosp1
    networks:
      - hospital
```

## Appendix B. Orderer Setup

```yaml
ca_orderer:
  image: hyperledger/fabric-ca:$IMAGE_TAG
  environment:
    - FABRIC_CA_HOME=/etc/hyperledger/fabric-ca-server
    - FABRIC_CA_SERVER_CA_NAME=ca-orderer
    - FABRIC_CA_SERVER_TLS_ENABLED=true
    - FABRIC_CA_SERVER_PORT=9054
  ports:
    - "9054:9054"
  command: sh -c 'fabric-ca-server start -b admin:adminpw -d'
  volumes:
    - ../organizations/fabric-ca/ordererOrg:/etc/hyperledger/fabric-ca-server
  container_name: ca_orderer
  networks:
    - hospital
```

## Appendix C. Create Channel

```
createChannel() {
        setGlobals 1
        # Poll in case the raft leader is not set yet
        local rc=1
        local COUNTER=1
        while [ $rc -ne 0 -a $COUNTER -lt $MAX_RETRY ] ; do
                sleep $DELAY
                set -x
                peer channel create -o localhost:7050 -c $CHANNEL_NAME --ordererTLSHostnameOverride orderer.lithium.com -f ./channel-artifacts/${CHANNEL_NAME}.tx
                res=$?
                { set +x; } 2>/dev/null
                let rc=$res
                COUNTER=$(expr $COUNTER + 1)
        done
        cat log.txt
        verifyResult $res "Channel creation failed"
        successln "Channel '$CHANNEL_NAME' created"
}
```

## Appendix D. Deploy Chaincode

```
CHANNEL_NAME=${1:-"hospitalchannel"}
CC_NAME=${2:-"patient"}
CC_SRC_PATH=${3:-"NA"}
CC_SRC_LANGUAGE=${4:-"javascript"}
CC_VERSION=${5:-"1.0"}
CC_SEQUENCE=${6:-"1"}
CC_INIT_FCN=${7:-"initLedger"}
CC_END_POLICY=${8:-"NA"}
CC_COLL_CONFIG=${9:-"../../private-collections/private-collections.json"}
DELAY=${10:-"3"}
MAX_RETRY=${11:-"5"}
VERBOSE=${12:-"false"}
HOSP3=${13:-"false"}
```

## Appendix E. Enroll Server

```
/**
 * @description Enrolls and registers the patients in the initLedger as users.
 */
async function initLedger() {
  try {
    const jsonString = fs.readFileSync('../patient-asset-transfer/chaincode/lib/initLedger.json');
    const patients = JSON.parse(jsonString);
    let i = 0;
    for (i = 0; i < patients.length; i++) {
      const attr = {firstName: patients[i].firstName, lastName: patients[i].lastName, role: 'patient'};
      await enrollRegisterUser('1', 'PID'+i, JSON.stringify(attr));
    }
  } catch (err) {
    console.log(err);
  }
}
```

## Appendix F. Create Patient in a Server

```javascript
async createPatient(ctx, patientId, firstName, lastName, age, address) {
    const exists = await this.patientExists(ctx, patientId);
    if (exists) {
        throw new Error(`The patient ${patientId} already exists`);
    }
    const patient = {
        firstName,
        lastName,
        docType: 'patient',
        age,
        address,
    };
    const buffer = Buffer.from(JSON.stringify(patient));
    await ctx.stub.putState(patientId, buffer);
}
```

## Appendix G. Create Doctor

```javascript
/**
 * @description Create doctors in both organizations based on the initDoctors JSON
 */
async function enrollAndRegisterDoctors() {
  try {
    const jsonString = fs.readFileSync('./initDoctors.json');
    const doctors = JSON.parse(jsonString);
    for (let i = 0; i < doctors.length; i++) {
      const attr = {firstName: doctors[i].firstName, lastName: doctors[i].lastName, role: 'doctor', speciality: doctors[i].speciality};
      // Create a redis client and add the doctor to redis
      doctors[i].hospitalId = parseInt(doctors[i].hospitalId);
      const redisClient = createRedisClient(doctors[i].hospitalId);
      (await redisClient).SET('HOSP' + doctors[i].hospitalId + '-' + 'DOC' + i, 'password');
      await enrollRegisterUser(doctors[i].hospitalId, 'HOSP' + doctors[i].hospitalId + '-' + 'DOC' + i, JSON.stringify(attr));
      (await redisClient).QUIT();
    }
  } catch (error) {
    console.log(error);
  }
};
```

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
