# Peer review of "A Framework for User-Focused Electronic Health Record System Leveraging Hyperledger Fabric"

_information, doi:10.3390/info14010051_

Round 1

Reviewer 1 Report

The study entitled: A Framework for User-Focused Electronic Health Record System Leveraging Hyperledger Fabric investigated the excellent problem. The study suggested a suitable mechanism for an electronic digital healthcare system. This research additionally outlines the difficulties encountered and problems that need to be resolved before Hyperledger Fabric can be successfully implemented in healthcare systems. Considering all types of blockchains available, the needs are met by Hyperledger Fabric,  which offers a distributed and secure environment for healthcare systems. Blockchain has the potential to transform healthcare by putting the patient at the center of the system and enhancing health data protection and interoperability. Also, by using grant and revoke access mechanisms, patients have complete control over their medical information and authorized doctors who  are allowed to view records. This functionality is made possible by the chaincode defined in the blockchain platform. The research study has both practitioner and research implications for the de velopment of secure blockchain-based EMRs. The over manuscript is good and well. However, a few things must be improved based on the given suggestions.

1. Paper looks like an engineering, not a research article. Therefore, the authors must analyze the results and compare the proposed work with the existing blockchain schemes. 

2. The healthcare data is not clear  in the manuscript, upload the data on the github and give the data features in the manuscript

3. The source code should be shared; public authors are doing research on the public blockchain.

4. Design the simulation environment and define the variables and parameters of the simulations.

5. The manuscript must be rewritten from abstract to conclusion to convert this work into research work.

6. The resources, computing capability, block sizes, and resource leakage are not defined by blockchain.

Reviewer 2 Report

The topic of this paper is interesting.

Some feedback on further improvement of the paper is given below:

·         The introduction section should clearly specify the contribution of this work.

·         Section 2 (literature review) section provides discussion on some of the existing work on Electronic Health Records, Blockchain and the adoption of Blockchain technologies in the Healthcare sector. However, it would be good to include a subsection on related work that focuses on “User-Focused” Electronic Health Record system using Blockchain.

·         In section 2, some detailed analysis of how this work is  different from others could be included.

·         Some of the figures (such as Figures 2, 9) are blurry. It would be good to update those figures.

·         Performance analysis (using different performance metrics) of the proposed system (from the implementation work) should have been included. The discussion should also include the scalability aspect of the proposed system.

Round 2

Reviewer 1 Report

The current version of manuscript is address my all comments. I'm satisfied with changes in the current version 

Author Response

Thank you very much for your comments.  We have submitted an amended manuscript addressing the comments of Reviewer B.  Please find attached.

Reviewer 2 Report

The changes are not highlighted in the uploaded manuscript and the response letter does not provide the specific details of the changes being made in the manuscript either. So, it is difficult to verify if the required changes have been made as the updated manuscript seems to be very similar to the original submission. 

I suggest the authors clarify/highlight the changes being made in the updated version.

Round 3

Reviewer 2 Report

Changes have been made in the updated manuscript.